# *Never Train from Scratch*: FAIR COMPARISON OF LONG-SEQUENCE MODELS REQUIRES DATA-DRIVEN PRIORS

**Ido Amos**
Tel Aviv University*

**Jonathan Berant**
Tel Aviv University

**Ankit Gupta**
IBM Research

## ABSTRACT

Modeling long-range dependencies across sequences is a longstanding goal in machine learning and has led to architectures, such as state space models, that dramatically outperform Transformers on long sequences. However, these impressive empirical gains have been by and large demonstrated on benchmarks (e.g. Long Range Arena), where models are randomly initialized and trained to predict a target label from an input sequence. In this work, we show that random initialization leads to gross overestimation of the differences between architectures and that pretraining with standard denoising objectives, using *only the downstream task data*, leads to dramatic gains across multiple architectures and to very small gaps between Transformers and state space models (SSMs). In stark contrast to prior works, we find vanilla Transformers to match the performance of S4 on Long Range Arena when properly pretrained, and we improve the best reported results of SSMs on the PathX-256 task by 20 absolute points. Subsequently, we analyze the utility of previously-proposed structured parameterizations for SSMs and show they become mostly redundant in the presence of data-driven initialization obtained through pretraining. Our work shows that, when evaluating different architectures on supervised tasks, incorporation of data-driven priors via pretraining is essential for reliable performance estimation, and can be done efficiently.

## 1 INTRODUCTION

Self-supervised pretraining is now widespread across most areas of machine learning, including NLP, speech, and vision (Touvron et al., 2023; Baevski et al., 2020; Reed et al., 2022). Given a downstream task, it is standard to finetune a pretrained model rather than train "from scratch", to achieve better performance (Raffel et al., 2019). Conversely, when developing new architectures with better inductive biases for particular skills, for example, for capturing long-range dependencies or for better algorithmic reasoning, it is still common to train on the task data from scratch with random initialization (Tay et al., 2020a; Delétang et al., 2022; Velivckovi'c et al., 2022; Dwivedi et al., 2022). This difference in practice stems not only from the computational overhead required for pretraining on massive datasets, but also to decouple the effects of the pretraining data and allow an apples-to-apples comparison, which would otherwise require a "standard" pretraining corpus for each scenario.

A prime example of the latter scenario is estimating capabilities in modeling long range dependencies in sequences, a setting where Transformers have reported inadequate performance on benchmarks designed as stress tests, such as Long Range Arena (LRA) (Tay et al., 2020a). This inefficacy of Transformers has led to a line of new architectures, suggesting changes to RNNs, CNNs and Transformers themselves, biasing them towards capturing long range dependencies, and achieving impressive performance on LRA, when trained from scratch (Gu et al., 2022a; Gupta et al., 2022a; Li et al., 2022; Ma et al., 2022). However, these results do not align with performance of pretrained Transformers ("foundation models"), that have displayed remarkable performance on tasks involving modeling long range dependencies, such as text summarization, code completion and protein folding, (Touvron et al., 2023; Jumper et al., 2021). Despite the significant progress in long sequence modeling, the reasons for sub-par performance of Transformers on long sequence benchmarks, such as LRA,

---

*{idoamos@mail.tau.ac.il, joberant@cs.tau.ac.il, ankitgupta.iitkanpur@gmail.com}.

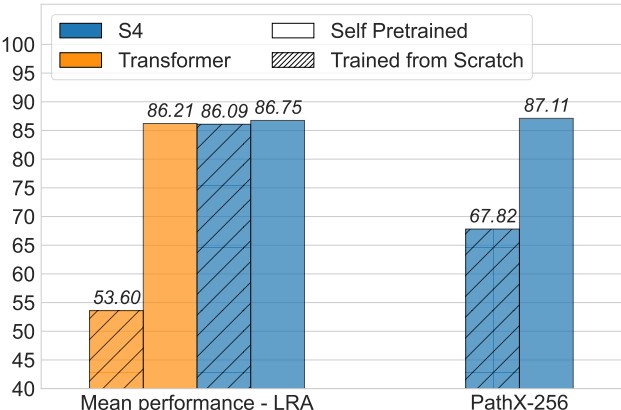

Figure 1: Evaluation of Transformers and S4 on Long Range Arena when trained from scratch vs. when self pretrained.

remains unexplored, while methods achieving competitive performance resort to tailored changes to the architecture (Ma et al., 2022; Zuo et al., 2022).

In this work, we shed light on this discrepancy, showing it stems from inadequate training and evaluation practices, and suggest a simple and efficient solution allowing a proper evaluation. While avoiding pretraining on a large corpus is understandable, training from a random initialization with downstream supervision alone disregards the role of the pretraining objective itself, leading to a different inductive bias than that of a pretrained model. In a recent line of work, El-Nouby et al. (2021); He et al. (2022); Krishna et al. (2023) have demonstrated that, when using denoising objectives, pretraining solely on downstream training data (denoted as *self pretraining*) often leads to gains comparable to the ones from pretraining on large corpora, showing effectiveness on tasks such as image classification, segmentation, text classification, etc. This suggests that, rather then training from scratch, a more realistic estimate of model performance can be obtained via self pretraining (SPT), with SPT acting as a data-driven initialization method, while allowing a fair comparison between methods as only the task data is used.

To demonstrate the importance of the suggested method, we empirically show that priors learned through SPT with denoising objectives are highly effective for learning long range dependencies across several architectures, eliminating the need for complex hand-crafted modeling biases used in current solutions (Gu et al., 2022a; Ma et al., 2022; Li et al., 2022; Orvieto et al., 2023). We primarily study Long Range Arena (LRA), a standard benchmark for long sequence modeling and evaluate multiple SPT models. We show that SPT improves the mean absolute performance of vanilla Transformers by more than 30%, for the first time allowing them to match the state-of-the-art performance on LRA without any architectural changes (Figure 1). This is in stark contrast to prior works where Transformers report significantly lower performance compared to the state-of-the-art.

We study the effectiveness of SPT for State Space models (SSMs), a novel line of architectures using modified linear RNNs as a replacement for attention layers in Transformers. Incorporating a specialized parameterization and initialization of linear RNNs, SSMs such as S4 successfully mitigate the vanishing/exploding gradient issues and reach impressive performance on long sequence tasks, such as LRA (Gu et al., 2022a). We find SPT to also benefit S4 with performance gains in 5 out of 6 LRA tasks. Moreover, with SPT, S4 solves the challenging PathX-256 task, achieving a 20% accuracy improvement compared to training from scratch (Figure 1). Building on these improvements, we study the utility of hand-crafted modeling biases in S4 over simpler linear RNNs, finding that the data-driven priors learned via SPT render most of them redundant (Gupta et al., 2022b). In doing so, we are the first to provide competitive performance with diagonal linear RNNs without any manual modifications (Orvieto et al., 2023).

Our findings show that priors beneficial for capturing distant dependencies can be simply learned from the task data via standard denoising objectives without any intrusive changes to the model. We examine the benefits of SPT across multiple data scales showing them to become even more pronounced as data becomes relatively scarce. Last, for SSMs, we analyze the convolution kernels

learned via SPT to shed light on the learned priors for capturing long-range dependencies. We demonstrate an interesting phenomena in which, depending on the modality, rapidly decaying kernels can lead to improved performance over the slowly decaying ones as used in the native S4 model, further highlighting the utility of learning priors from the data itself (Gu et al., 2020).

Our main contributions can be summarized as follows:

*(i)* We show that the reported performance of various architectures on long range benchmarks is grossly underestimated, and suggest an inexpensive data-driven approach to enable accurate evaluation without requiring any additional data.

*(ii)* We report large empirical gains over the previously-reported performances on LRA across a range of architectures and, in particular, improve upon the best reported accuracy on the challenging PathX-256 task by 20 absolute points ($67 \rightarrow 87$).

*(iii)* We demonstrate how manually-designed biases become increasingly redundant with pretraining and that, with modern training and evaluation practices, simpler models can often match the performance of sophisticated architectures. We are the first to provide competitive performance on LRA with Transformers and diagonal linear RNNs.

The multi-modal and challenging setup of LRA, along with the scale of improvements due to SPT, advocate the inclusion of a pretraining stage while evaluating models in general, for example when designing architectures for multidimensional inputs (Nguyen et al., 2022), algorithmic reasoning (Diao & Loynd, 2023) or graphs (Shirzad et al., 2023).

Our code & data are available at `https://github.com/IdoAmos/not-from-scratch`.

## 2 EXPERIMENTAL SETUP

Our experiments center around the evaluation of Transformers and SSMs on the Long Range Arena (LRA) benchmark which was proposed for examining the ability of sequence models to capture long-range dependencies (Tay et al., 2020a). It contains 6 main sequence classification tasks, each being either binary or 10-way sequence classification.

1. ListOps: Each sequence in the dataset is a nested list, with each sublist describing an operation (e.g. MAX, MEAN) to be applied on a set of tokens (Nangia & Bowman, 2018). Evaluation of nested lists are used as a single token in their enclosing list thus requiring the understanding of hierarchical structure, the task is 10-way classification with sequence length of $2K$.

   **INPUT:** `[MAX 4 3[MIN 2 3]1 0[MEDIAN 1 5 8 9 2]]` **OUTPUT:** 5

2. Text: a character-level version of the IMDb reviews dataset (Maas et al., 2011) for sentiment classification, the task is binary classification with sequence length of up to $2048$.

3. Retrieval: a character-level version of the AAN dataset (Radev et al., 2013) for predicting similarity scores of two documents. The task is binary classification with sequence length of up to $4K$, requiring to process $8K$ tokens for evaluation.

4. Image: grayscale CIFAR10 images are flattened as 1D sequences and any explicit 2D inductive bias cannot be used. The task is 10-way classification, with sequence length $1024$.

5. Pathfinder, PathX: synthetic 2D visual tasks treated as 1D sequences (similar to Image) for testing tracing capabilities (Linsley et al., 2018; Kim et al., 2020). PathX and Pathfinder are similar tasks that differ in sequence length (1024 vs 16384) and are binary classification.

Apart from the aforementioned tasks, we examine an additional variant of PathX called PathX-256 with sequence length $256^2 = 65536$ and we are the first to report strong results on this task. Besides LRA, we experiment with additional datasets that will be described later in Section 3.7.

**Self Pretraining (SPT)**   We perform SPT with a causal/autoregressive sequence modeling objective for unidirectional models, and a masked sequence modeling objective for bidirectional models, using *only* the downstream task training set. For the visual tasks (Image, Pathfinder, Path-X) the masking ratio for masked sequence modeling is set to 50% following He et al. (2022), to 15% for language

Table 1: **Long Range Arena**. (top) performance of models trained from scratch as reported in Tay et al. (2020a), (bottom) performance of self pretrained (SPT) Transformers of sizes *comparable* to the ones on top. ✗ denotes chance accuracy.

| Approach | Listops | Text | Retrieval | Image | Pathfinder | PathX | Avg. |
|---|---|---|---|---|---|---|---|
| Transformer | 36.37 | 64.27 | 57.46 | 42.44 | 71.40 | ✗ | 53.66 |
| Local Attention | 15.82 | 52.98 | 53.39 | 41.46 | 66.63 | ✗ | 46.71 |
| Longformer | 35.63 | 62.85 | 56.89 | 42.22 | 69.71 | ✗ | 52.88 |
| Linformer | 35.70 | 53.94 | 52.27 | 38.56 | 76.34 | ✗ | 51.14 |
| Reformer | 37.27 | 56.10 | 53.40 | 38.07 | 68.50 | ✗ | 50.56 |
| BigBird | 36.05 | 64.02 | 59.29 | 40.83 | 74.87 | ✗ | 54.17 |
| Linear Trans. | 16.13 | 65.90 | 53.09 | 42.34 | 75.30 | ✗ | 50.46 |
| Performer | 18.01 | 65.40 | 53.82 | 42.77 | 77.05 | ✗ | 51.18 |
| Transformers + Masked SPT | **59.75** | **89.27** | 88.64 | 74.22 | 88.45 | 87.73 | 81.34 |
| Transformers + Causal SPT | 59.15 | 88.81 | **90.38** | **76.00** | **88.49** | **88.05** | **81.81** |

tasks (Text, Retrieval) following Liu et al. (2019), and to 10% for ListOps. For Transformers, we use full attention as default with the hardware-optimized FLASH implementation (Dao et al., 2022). Due to computational constraints, for tasks with sequence length at least $16K$ we split the input to the attention layer into non-overlapping blocks of size $4096$ and allow each block to attend to itself and its neighbour(s).

Our codebase is built on the original S4 repository.[1] For additional experimental details, such as computational resources for SPT and finetuning, please refer to Appendix C.1.

## 3 RESULTS

In Section 3.1, we perform SPT for LRA tasks using the official model configurations. In Section 3.2, we perform SPT for Transformers and S4. Section 3.3 evaluates the role of design choices in SSMs in the context of SPT. Section 3.4 examines the utility of SPT across data scales and Section 3.5 examines the utility of PT on a large text corpus. Section 3.6 provides an analysis of pretrained SSM kernels and how they relate to current initialization schemes. Section 3.7 contains additional experiments on distinct modalities.

### 3.1 UNDERESTIMATION OF LONG-RANGE ABILITIES OF TRANSFORMERS

We start by investigating the reliability of the historically-reported model performances on LRA, in the more modern setting of pretraining. Concretely, we repeat the Transformer experiments performed by Tay et al. (2020a), except that we first pretrain the model on the task data and then finetune it. To allow fair comparison with the original results, we strictly follow the model configurations used by Tay et al. (2020a). We experiment with two pretraining objectives: (1) next token prediction for unidirectional models (2) masked token prediction for bidirectional models, varying the masking ratio as detailed in Section 2.

As summarized in Table 1, we find that both pretraining objectives lead to dramatic performance gains for Transformers compared to the conventional practice of training with random initialization, with the average test accuracy increasing by roughly $30\%$. Both causal and masked pretraining yield similar results even in cases where there are no clear benefits to using a causal model, such as on the visual tasks. Furthermore, even for LISTOPS large performance gains are observed even though, in the original data, the arguments to the list operations are sampled randomly, meaning that inferring missing tokens from the context is rarely possible.

As the experiments are performed with no architectural changes or additional data, the difference in performances can be attributed to the priors learned during SPT, clearly demonstrating its importance for a reliable performance evaluation.

---

[1]`https://github.com/HazyResearch/state-spaces`

Table 2: **Long Range Arena.** Self pretrained (SPT) Transformers and S4 compared to existing trained from scratch models. Average performance ("Avg.") is reported without PathX-256 to align with prior work. Results for MEGA, SPADE & S4 are taken from original papers with exceptions denoted by †. ✗ denotes computationally infeasible, ❏ denotes unreported results.

| Approach | Listops | Text | Retrieval | Image | Pathfinder | PathX | PathX-256 | Avg. |
|---|---|---|---|---|---|---|---|---|
| Transformers + Rotary | 47.90 | 79.08 | 82.31 | 75.04 | 76.64 | 84.72 | ✗ | 74.28 |
| Transformers + Rotary + Masked SPT | 61.49 | **91.02** | **91.57** | 86.04 | 94.16 | 92.98 | ✗ | 86.21 |
| S4 (Gu et al., 2022a) | 59.60 | 86.82 | 90.90 | 88.65 | 94.20 | 96.35 | 67.82† | 86.09 |
| S4 + Masked SPT | 61.25 | 90.34 | 88.74 | 89.36 | 94.92 | 96.94 | **87.11** | 86.75 |
| SPADE (Zuo et al., 2022) | 60.50 | 90.69 | 91.17 | 88.22 | **96.23** | 97.60 | ❏ | 87.40 |
| MEGA (Ma et al., 2022) | **63.14** | 90.43 | 91.25 | **90.44** | 96.01 | **97.98** | ❏ | **88.21** |
| Pythia 70M (Rand Init) | 41.20 | 69.29 | 76.45 | 52.55 | 74.31 | ✗ | ✗ | 62.76 |
| Pythia 70M | 43.05 | 83.41 | 84.29 | 67.41 | 80.05 | ✗ | ✗ | 68.04 |

## 3.2 COMPARING S4 AND TRANSFORMERS

In the above set-up we strictly adhered to the model sizes used by Tay et al. (2020a) and consequently the absolute performances are still low compared to the current state-of-the-art on LRA. In this section, we scale the model sizes and evaluate the utility of SPT for the best performing architectures including S4 (Gu et al., 2022a). For Transformers, we replace the positional embeddings with the more commonly used rotary embeddings (Su et al., 2021) and only train bidirectional models in line with prior works reporting high performance.

As summarized in Table 2, SPT leads to dramatic performance gains for Transformers with performance gains ranging from $8 - 15\%$ across tasks, even surpassing the average performance of a well-tuned S4 ($86.2$ vs $86.1$). SPT Transformers surpass the performance of both trained from scratch and SPT versions of S4 on 3 out of 6 tasks. The results in Table 2 defy current understanding, with prior works citing the sub-par LRA performance of Transformers as a prime motivating factor for new methods. Yet we show that, while architectural developments indeed lead to remarkable performance gains, most of the priors essential to high performance can already be learned from data directly.

In case of S4, while SPT leads to modest gains on most tasks, a substantial gain of $20\%$ is observed on the challenging PathX-256 task with input length of $65K$, significantly improving over the best reported performance of $63.1\%$ by (Dao et al., 2022) who, in addition, used extra data from the Pathfinder-64 task.

The additionally reported models, SPADE and MEGA, are Transformer variants that augment the model with a single or several state space layers. SPADE combines the outputs of a frozen S4 layer and local attention in the first block, while MEGA incorporates a learned exponential moving average, an instance of diagonal SSMs, into gated attention blocks. To the best of our knowledge, we are the first to show that purely attention-based methods, without any architectural modifications, can achieve competitive results on LRA. While incorporating SSMs can be important in terms of scalability to longer sequences due to their log-linear complexity with respect to input length, we show that in terms of model performance, pretraining leads to biases that are as effective as manual designs.

An important aspect of SPT is the use of additional compute compared to the trained from scratch baseline and it is natural to investigate if similar gains can be obtained by training from scratch for longer. For all our trained from scratch baselines, we ensured that the validation performance had converged and did not improve for several consecutive epochs. We examine the aspect of the computational overhead of SPT in detail Appendix D, where we show that SPT leads to significant gains, even in the setting where the same amount of compute is used for SPT models and the ones that are trained from scratch.

## 3.3 THE ROLE OF EXPLICIT PRIORS

We have established that SPT allows for a more reliable evaluation of the actual capabilities of architectures and further improves the performance of SSMs such as S4. Despite its high performance, S4 has a complex design guided by principled theoretical considerations to enable long range signal

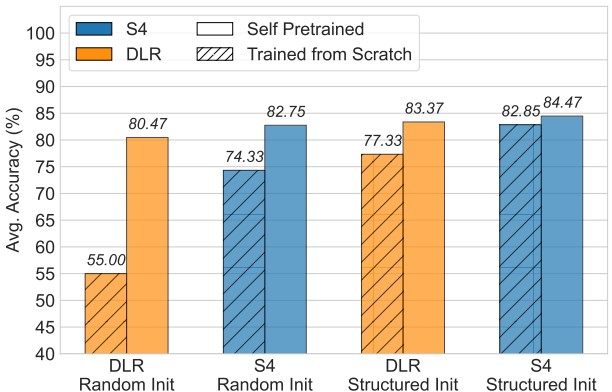

Figure 2: Average performance of models when trained from scratch or self pretrained, for different sets of initializations prior to pretraining. See Table 7 for per-task results.

propagation, which can explain the small advantage maintained over SPT Transformers, lacking such an inductive bias. In a series of works, various simplifications to S4 have been proposed while maintaining performance. We will now show that SPT allows for an even simpler model (viz. diagonal linear RNN) to match the performance of S4.

We first provide a brief overview of SSMs below and refer to Gu et al. (2022a) for a detailed description. Given an input scalar sequence[2] $u$, SSMs follow a linear recurrence generating a hidden state vector $\vec{x}_n$ at timestep $n$, and produce a scalar output sequence $y$ as

$$
\begin{aligned}
\vec{x}_n &= \boldsymbol{A}\vec{x}_{n-1} + \boldsymbol{B}u_n & \boldsymbol{A} \in \mathbb{C}^{N \times N}, \boldsymbol{B} \in \mathbb{C}^{N \times 1} \\
y_n &= \boldsymbol{C}\vec{x}_n & \boldsymbol{C} \in \mathbb{C}^{1 \times N}
\end{aligned}
\tag{1}
$$

By unrolling the recurrence across the timesteps, it can be shown that $y$ can be equivalently computed by convolving $u$ with the kernel defined by $K_k = \boldsymbol{C}^T \boldsymbol{A}^k \boldsymbol{B}$. Instead of directly using $\boldsymbol{A}$, $\boldsymbol{B}$, $\boldsymbol{C}$ as learnable parameters, S4 uses an alternate parameterization inspired by a theory in continuous time, motivating the transformations:

$$
\boldsymbol{A} = \boldsymbol{\Lambda} - \boldsymbol{P}\boldsymbol{Q}^* \tag{2.1}
$$

$$
\bar{\boldsymbol{A}} = (\boldsymbol{I} - \Delta/2 \cdot \boldsymbol{A})^{-1}(\boldsymbol{I} + \Delta/2 \cdot \boldsymbol{A}) \tag{2.2}
$$

$$
\bar{\boldsymbol{B}} = (\boldsymbol{I} - \Delta/2 \cdot \boldsymbol{A})^{-1}\Delta\boldsymbol{B} \quad \bar{\boldsymbol{C}} = \boldsymbol{C} \tag{2.3}
$$

$$
\boldsymbol{K}_k = \bar{\boldsymbol{C}}^T \bar{\boldsymbol{A}}^k \bar{\boldsymbol{B}} \tag{2.4}
$$

where $\boldsymbol{\Lambda}, \boldsymbol{P}, \boldsymbol{Q}, \boldsymbol{B}, \boldsymbol{C}, \Delta$ are learnable parameters and $\boldsymbol{\Lambda} \in Diag(\mathbb{C}^{N \times N})$, $\boldsymbol{P}, \boldsymbol{Q} \in \mathbb{C}^{N \times 1}$. In addition to this parameterization, S4 uses a principled initialization method aimed towards a slow decay of the kernel (w.r.t. timestep $k$) in order to facilitate capturing long-range dependencies.

Inspired by the success of S4, Gupta et al. (2022b) proposed a simplification to S4 called Diagonal Linear RNN (DLR) defined as

$$
\begin{aligned}
\vec{x}_n &= \boldsymbol{\Lambda}\vec{x}_{n-1} + \boldsymbol{1}u_n & \boldsymbol{\Lambda} \in diag(\mathbb{C}^{N \times N}) \\
y_n &= \boldsymbol{C}\vec{x}_n & \boldsymbol{C} \in \mathbb{C}^{1 \times N}
\end{aligned}
\tag{3}
$$

where $\boldsymbol{1}$ is the all-ones vector. DLR is significantly simpler to compute compared to S4 and the authors reported it to be as performant as state-of-the-art SSMs on a wide variety of token-level tasks. Hence, it is natural to investigate the conditions under which S4 with its more complex design (eq. 2) can be replaced by the simpler DLR. To that end, we evaluate the performance of DLR and S4 on ListOps, Text, Image and PathX tasks as they are the hardest and represent all modalities in LRA. For each model, we experiment with two sets of initializations: (1) random initialization where the state space parameters are initialized from a normal distribution with a small standard deviation, and

---

[2]When the input is a sequence of vectors, the model is applied to each channel separately and is commonly followed by a FFN to exchange information across channels.

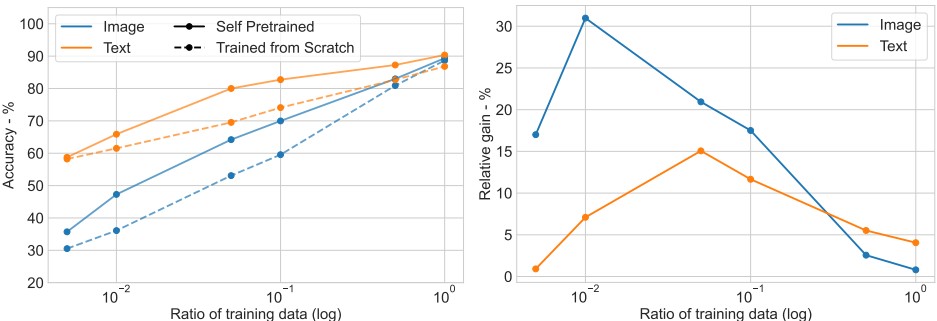

Figure 3: Trained from scratch and self pretrained (SPT) versions of S4 evaluated on multiple data scales for Image and Text tasks from LRA, originally containing $45K$ and $25K$ samples respectively. (left) absolute performances and (right) relative gains due to SPT over training from scratch.

(2) "structured" initialization recommended by the respective authors aimed at capturing long-range dependencies.

The results are summarized in Figure 2 and per-task results are provided in Table 7. We find that, when trained from scratch, with both random and structured initializations, DLR lags behind S4 in terms of average performance (77 vs 83) demonstrating that biases incorporated through the specific initialization and parameterization used in S4 are indeed critical to performance. However, the picture radically changes under SPT – with SPT, DLR outperforms a trained from scratch S4 (83.4 vs 82.8) and is only slightly behind SPT S4 (83.4 vs 84.5). This suggests that the data-driven priors learned through pretraining are almost as effective as the manual biases incorporated in S4.

Results in this section have two additional implications. First, this is the first instance in which vanilla diagonal linear RNNs have been shown to achieve competitive performance on LRA. Prior work by Orvieto et al. (2023) suggested an additional normalization step in the kernel generation on top of a tailor-made initialization to achieve high performance on LRA. Second, while our discussion revolved around SSMs, many subsequent works on designing global convolutions followed similar principles. For example, Li et al. (2022) proposed to generate a decaying convolution kernel from shorter kernels via interpolation, which induces smoothness and can be viewed as a normalization step. Similarly, Fu et al. (2023) applied a global convolution layer that is transformed by a deterministic function to explicitly induce a smoother kernel. Yet our results suggest that these explicit steps are less significant when models are self pretrained.

## 3.4 Self pretraining is Effective Across Data Scales

As the priors learned via SPT are data-driven, their efficacy is dependent on the training set itself, which leads us to examine the performance gains as a function of the dataset size. To this end, given a downstream task, we randomly sample a subset of the training set, and study the performance gains for S4 due to SPT under varying sizes of the subset. We restrict the pretraining phase of S4 to a fixed number of update steps across all experiments and finetune until convergence.

As summarized in Figure 3, we uncover an interesting phenomenon; while the relative gains from SPT over the trained from scratch baseline S4 are modest when the full task data is available, they become increasingly significant (and as large as $30\%$) on smaller data scales. This shows that priors from pretraining are especially effective when training data is scarce and, in the context of previous sections, implies that the incorporation of the pretraining stage is important for model evaluation regardless of dataset size. In Appendix E, we provide a complementary study on the effectiveness of SPT across model sizes, demonstrating that indeed SPT is effective across multiple model scales for both S4 and Transformers.

## 3.5 pretraining on text corpora

Given the widespread success of pretrained language models and the large gains due to SPT on the LRA tasks (Table 2), it is natural to ask if similar gains could be achieved by finetuning a language

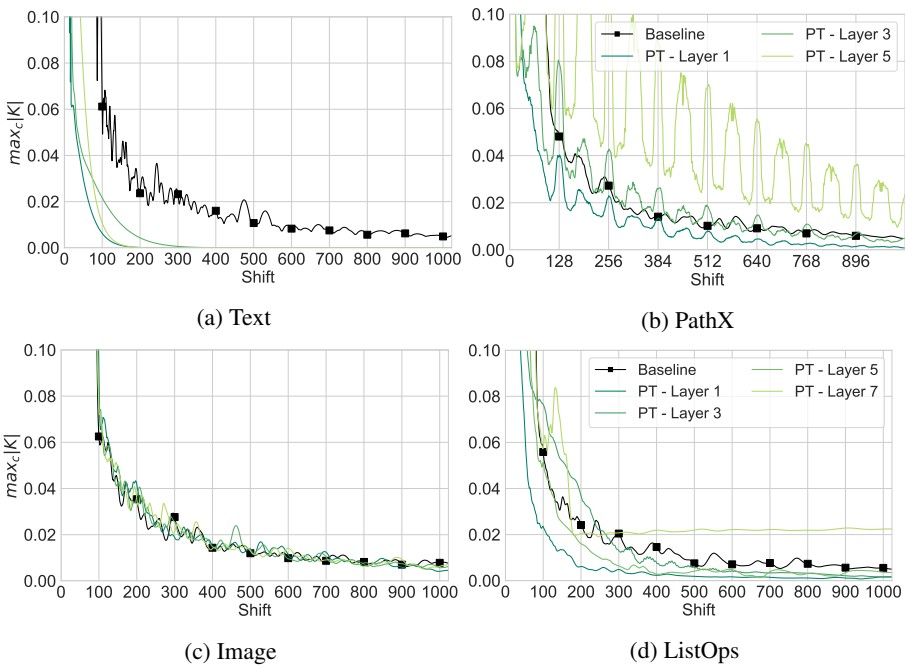

Figure 4: Maximal absolute values of kernels across channels in S4 learned via self pretraining (PT) compared against the standard HiPPO kernels (Baseline). Only odd layers are shown for better visualization.

model pretrained on a large text corpus. To answer this, we consider Pythia 70M (Biderman et al., 2023), an autoregressive Transformer pretrained on the Pile (Gao et al., 2020) as well as a randomly initialized version with the same architecture, denoted as "Pythia 70M (RandInit)" in Table 2. To be comparable to existing results and due to the formal requirements of the LRA benchmark, we use character/pixel-level tokenization instead of the original BPE tokenizer and the model is required to adapt to the new tokenization during finetuning.

As shown in Table 2, Pythia 70M generally lags behind our trained from scratch Transformer baseline due to the changed tokenization and difference between the pretraining distribution and the downstream tasks. This further highlights the importance of SPT as it allows the model to specifically learn and adapt to the structure and modality of the given task data. However, the performance of Pythia 70M is significantly better that its randomly initialized version Pythia 70M (Rand init) suggesting that pretraining on large text corpora can be beneficial across modalities.

### 3.6 THEORETICALLY-DERIVED VS DATA-DRIVEN KERNELS

Many high performing models such as SSMs incorporate manually-crafted priors to bias the model towards learning long range dependencies. For example, the initializations used in SSMs such as S4, DSS, S4D, S5 are based on HiPPO theory (Gu et al., 2020), which explicitly determines the decay rate of the convolution kernels over time and provides strong dependence between distant elements in the input sequence. In similar spirit, Li et al. (2022) generate convolution kernels modified with fixed weights aimed towards a slow decay. On the other hand, kernels learned via SPT have no guarantees of a slow decay and depend solely on the input distribution and the pretraining objective.

In this section, we analyze the structure of the convolutional kernels learned via SPT and compare them to the HiPPO-based kernels used to initialize existing SSMs such as S4. The convolution operation in S4 has the form

$$y_{c,k} = \sum_{l=0}^{k} \bar{C}_c^T \bar{A}_c^l \bar{B}_c x_{c,k-l} = \sum_{l=0}^{k} K_{c,l} \cdot x_{c,k-l} \tag{4}$$

where $c$ is the channel and $k$ is the timestep. Based on this structure, we can estimate the degree of dependence between sequence elements at channel $c$, $l$ positions apart as $|K_{c,l}|$. For easier

Table 3: **Additional Experiments.** Performance on Speech Commands (SC), sCIFAR (accuracy) and BIDMC (R2) tasks. Results for trained from scratch S4 taken from Gu et al. (2022a), except for BIDMC (denoted by †) that are reproduced for the more interpretable R2 score.

| Approach | SC | | sCIFAR | BIDMC | | |
|---|---|---|---|---|---|---|
| | Causal | Bi. | | HR | RR | SpO2 |
| S4 | 93.60 | 96.08 | 91.13 | **0.999**$^\dagger$ | **0.994**$^\dagger$ | **0.999**$^\dagger$ |
| Transformers | 84.55 | 86.93 | 79.41 | 0.998 | 0.981 | 0.998 |
| S4 + SPT | **95.09** | **96.52** | **91.67** | 0.999 | 0.990 | 0.997 |
| Transformers + SPT | 86.13 | 91.49 | 90.29 | 0.992 | 0.956 | 0.993 |

interpretation, we take the maximal absolute value over the channels[3] as $K_{\max,l} = \max_c |K_{c,l}|$. For a shift $l$, $K_{\max,l}$ bounds the norm of the derivative of $y_{c,k}$ w.r.t $x_{c,k-l}$ for all positions $k$ and channels $c$.

We generate kernels for the pretrained S4 models from Section 3.2 (before finetuning) and compare with the ones used in standard S4. Figure 4 plots $K_{\max}$ for the Image, Text, PathX and ListOps, all entailing better performance with the pretrained model (Table 2). We observe that the learned kernels exhibit variable decay rates across the tasks and model layers, in contrast to the fixed decay rate of the data-agnostic HiPPO kernels. In particular, on the Text task the learned kernels are more local compared to HiPPO. For PathX, the vertical grid lines are aligned with the image resolution $(128 \times 128)$ showing high correlation between the underlying 2D structure of the data and the kernel peaks. Overall, Figure 4 further highlights the utility of SPT over data-agnostic initializations that cannot adapt to a local or global structure in a task distribution.

## 3.7    ADDITIONAL EXPERIMENTS

In addition to LRA, we also tested the utility of SPT on 3 additional datasets, encompassing 2 additional natural data modalities, described as follows:

- **Speech Commands (SC)**  Raw speech waveforms of length $16K$ used in a 35-way classification task (Warden, 2018). We test both causal and bidirectional models following Gu et al. (2022b).

- **sCIFAR**  Sequential CIFAR-10 dataset using RGB channels as features. This is similar to the Image task from LRA that uses grayscale images, except that here richer features are used.

- **BIDMC**  A suite of 3 regression tasks, requiring to predict respiratory rate (RR), heart rate (HR) and blood oxygen saturation (SpO2) from EKG and PPG signals of length $4K$ each.

The results shown in Table 3, with additional details in Appendix C.3, further strengthen the claims made throughout this work. On both SC and sCIFAR tasks, SPT leads to large performance gains for Transformers and modest gains for S4. The gaps between trained from scratch Transformer and S4 are substantially narrowed with SPT. On the SC task, SPT leads to a large $5\%$ improvement for Transformers and we observe the performance gap between causal and bidirectional variants of S4 to be mitigated with SPT. A similar, but not identical, observation is made in section 3.1 where masked and causal SPT lead to very similar results on all LRA tasks.

On the sCIFAR task, SPT leads to a dramatic $11\%$ improvement for Transformers, nearly matching the performance of S4 (90.3 vs 91.7) and again pointing towards a sub-optimal evaluation when only training from scratch. On BIDMC, the performances of both Transformer and S4 baselines are already close to perfect and it is hard to observe any meaningful improvements due to SPT.

In general, our results suggest that similar under-estimation of model performances might also be prevalent in other scenarios where training from scratch is standard (Delétang et al., 2022; Velivckovi'c et al., 2022; Dwivedi et al., 2022).

---

[3]since the model is bidirectional there are two sets of kernels, left to right, and right to left. We take the maximum over both.

## 4 ACKNOWLEDGMENTS

We thank Amir Globerson for insightful discussions and his support throughout the course of this work. This research was partially supported by The Yandex Initiative for Machine Learning, and the European Research Council (ERC) under the European Union Horizons 2020 research and innovation programme (grant ERC DELPHI 802800).

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

## A  Related Work

**Modeling Long Range Dependencies**  Evaluation of long-sequence models commonly includes the LRA benchmark (Tay et al., 2020a), a suite of tasks demonstrating the inefficacy of various efficient Transformers on long sequences,(Tay et al., 2020b). The first to obtain high performance on LRA was the S4 model (Gu et al., 2022a), an instance of linear RNNs augmented according to a complementary theory in continuous time (Gu et al., 2020; 2022b). Following S4, multiple works have proposed simplifications to S4 (Gupta et al., 2022a; Gu et al., 2022b; Smith et al., 2023; Orvieto et al., 2023) or augmentations to other common architectures (Li et al., 2022; Fu et al., 2023; Ma et al., 2022; Zuo et al., 2022), aimed at replicating biases observed in S4 and achieving similar performance on LRA. Common to all of the above is that evaluation on the tasks is done from a random initialization and thus does not encompass the biases from a various pretraining objectives, and specifically denoising. In Gupta et al. (2022b), the authors examine diagonal linear RNNs and Transformers on a set of long range tasks with dense (token-level) supervision but do not evaluate on LRA.

**Pretraining with Downstream Data**  Pretraining prior to evaluation is already the best practice in many domains and is usually performed using upstream corpora that are much larger than the intended downstream task. While various objectives could be used for pretraining (Lewis et al., 2020; Baevski et al., 2020; Assran et al., 2023) we focused on the typical causal and masked language modeling objectives (Radford et al., 2019; Liu et al., 2019). El-Nouby et al. (2021) were the first to show that self-supervised pretraining on the downstream task data alone can often match performance of supervised ImageNet pretraining, for object detection and segmentation tasks. In Krishna et al. (2023), the authors provide a similar study on NLP tasks, showing multiple instances in which pretraining on the task data performs comparably to off-the-shelf models pretrained on large text corpora.

## B  Conclusions

In this work, we argued against the common practice of training models from scratch to evaluate their performance on long range tasks and suggested an efficient and effective solution to mitigate this issue – self-supervised pretraining on the task data itself. Through a comprehensive array of experiments, we showed that our method consistently leads to dramatically improved performance for multiple architectures across data scales, and allows simpler models to match the performance of complex ones. Our work stresses the need to account for the pretraining stage while designing and evaluating novel architectures in the future.

## C  Appendix

### C.1  Additional Experiment Details & Hyper-Parameters

We provide additional implementation notes and hyper-parameters for reproduction purposes. For all experiments listed in the main text, pretraining is performed either with cross-entropy (CE) or L1 loss, and the normalization layer for Transformers always uses LayerNorm in the PreNorm setting. When finetuning a pretrained model we use the checkpoint matching highest validation accuracy / R2 score for CE / L1 loss respectively, and perform a search over a small grid of learning rates and batch sizes with log spacing, e.g. (1e-3 5e-4 1e-4) and (8 32 128), with values for best performance reported in Table C.1.

Our experiments were performed on NVIDIA 3090 and V100 GPUs. All models were trained for a maximum of either 200 epochs or 24h on a single GPU (whichever comes first) for pre-training and fine-tuning, with exceptions for Transformers on PathX & Speech Commands experiments that were trained for a maximum of 5 days on 4 GPUs, and Pythia experiments that were trained for a maximum of 2 days on 4 GPUs. When detailing learning rates for State Space models, learning rates for parameters of the State Space layer itself are set differently to predefined values, we always use values provided by respective authors (Gu et al., 2022a; Gupta et al., 2022b).

Additional specifications such as learning rate schedules and optimizers can be found in our repository.

### C.1.1 UNDERESTIMATION OF LONG-RANGE ABILITIES OF TRANSFORMERS

Model hyper-parameters (e.g. model size, num. layers, num. attention heads etc.) and model configuration (classification head, normalization layer type etc.) are similar to those listed in Appendix A in (Tay et al., 2020a) and provided here in Table 4, with the main exception being the pooling method across sequence dimension during finetuning. We use mean-pooling, max-pooling or the last element of the sequence, instead of using a `[CLS]` token, as done in Tay et al. (2020a). The choice of pooling method for SPT models had most impact on the PathX tasks, but varying across max/mean/last pooling with trained-from-scratch Transformers did not exceed random performance, same as Tay et al. (2020b).

### C.1.2 COMPARING S4 AND TRANSFORMERS

As listed in the main text, in this section there are discrepancies between model sizes of Transformers and S4. For Text & ListOps, the models in official repository of Gu et al. (2022a) are smaller then the Transformers used in LRA, we tried larger S4 models (from scratch and SPT) matching the Transformer in parameter count which did not result in better performance. For hyper-parameters of S4 models we refer to Gu et al. (2022a) - Appendix D.2 Table 11, and Table 5 for results of our grid searches. Hyper-parameters of Transformers can be found in Table 4. For the PathX-256 task we use the same configuration as PathX, changing the pooling method again from Mean to Max. For pretraining we use learning rate 1e-3, batch size 8 and batch size 64 for fine tuning with the same learning rate, the trained from scratch model's learning rate is 1e-4 with batch size 16.

### C.1.3 THE ROLE OF EXPLICIT PRIORS

S4 models follow the setup described in C.1.2 for structured and random initialization, with results for grid searches in Table 5. For DLR (Gupta et al., 2022b), we use the official implementation[4], and set the State Space layer parameter as the default, with state size 1024. For the remaining hyper-parameters (model size, num. layers, weight decay) we use the same values as S4, with the exception of the normalization layer that is fixed to LayerNorm. For the random initialization, we reset all model parameter value to random normal values and standard deviation 0.1. For all experiments on PathX we use Max-pooling on the final sequence representation, which we found helpful for optimization.

### C.1.4 SELF PRETRAINING IS EFFECTIVE ACROSS DATA SCALES

Experiments in this section use similar hyper-parameters to Table 5. For data restriction, we use fractions of the entire training set: (Num. Train Samples) $\times \{0.005, 0.01, 0.05, 0.1, 0.5, 1.0\}$.
The subsets are inclusive, meaning smaller fractions are contained in bigger ones, to avoid biases from difficult sampled subsets. For experiments on Text we pretrain all models for $40K$ pretraining steps and finetune the checkpoint with best performance on the validation set until convergence. For Image we pretrain all models for $10K$ step and finetune similarly. The learning rate and batch sizes are fixed across all subsample experiments, on Text we use batch size 8 and learning rate 5e-4 for pretraining and 16, 5e-4 for finetuning, for the trained from scratch model we use batch size 16 and learning rates 1e-3, 5e-4 and use the model with best performance on the validation set. For Image we use batch size 50 for all experiments, learning rate 1e-2 for pretraining, and learning rate to 2e-3 for finetuning and training from scratch.

---

[4] https://github.com/ag1988/dlr

Table 4: hyper-parameters for Transformers in all sections, section 3.1 (denoted by †), where 2 values are listed the 2nd value was used for pretraining, e.g. (FT, PT). LR - Learning Rate, BSZ - Batch Size, WD - Weight Decay

| ListOps | Features | Depth | Num Attn. Heads | FF size | Pooling | LR | BSZ | WD | PT Loss |
|---|---|---|---|---|---|---|---|---|---|
| Transformer + Masked SPT† | 512 | 6 | 8 | 1024 | Mean | 1e-4,1e-3 | 64,128 | 0.1 | CE |
| Transformer + Causal SPT† | 512 | 6 | 8 | 1024 | Last | 5e-4, 5e-4 | 128,32 | 0.1 | CE |
| Transformer + Rotary + Masked SPT | 512 | 6 | 8 | 1024 | Mean | 1e-4, 1e-3 | 64,256 | 0.05 | CE |
| **Text** | **Features** | **Depth** | **Num Attn. Heads** | **FF size** | **Pooling** | **LR** | **BSZ** | **WD** | **PT Loss** |
| Transformer + Masked SPT† | 512 | 6 | 8 | 1024 | Mean | 1e-4,5e-4 | 64,32 | 0.1 | CE |
| Transformer + Causal SPT† | 512 | 6 | 8 | 1024 | Mean | 1e-4,1e-3 | 64,32 | 0.1 | CE |
| Transformer + Rotary + Masked SPT | 512 | 6 | 8 | 1024 | Mean | 5e-4,5e-4 | 64,8 | 0.1 | CE |
| **Retrieval** | **Features** | **Depth** | **Num Attn. Heads** | **FF size** | **Pooling** | **LR** | **BSZ** | **WD** | **PT Loss** |
| Transformer + Masked SPT† | 128 | 4 | 4 | 512 | Mean | 5e-4,5e-3 | 16,32 | 0 | CE |
| Transformer + Causal SPT† | 128 | 4 | 4 | 512 | Mean | 1e-3,5e-3 | 48,32 | 0 | CE |
| Transformer + Rotary + Masked SPT | 128 | 6 | 4 | 512 | Mean | 5e-4,5e-3 | 16,32 | 0 | CE |
| **Image** | **Features** | **Depth** | **Num Attn. Heads** | **FF size** | **Pooling** | **LR** | **BSZ** | **WD** | **PT Loss** |
| Transformer + Masked SPT† | 64 | 3 | 4 | 128 | Max | 1e-3,1e03 | 16,32 | 0 | L1 |
| Transformer + Causal SPT† | 64 | 3 | 4 | 128 | Max | 1e-3,5e-3 | 32,32 | 0 | L1 |
| Transformer + Rotary + Masked SPT | 256 | 6 | 4 | 512 | Mean | 1e-3,1e-3 | 64,32 | 0 | L1 |
| **Pathfinder** | **Features** | **Depth** | **Num Attn. Heads** | **FF size** | **Pooling** | **LR** | **BSZ** | **WD** | **PT Loss** |
| Transformer + Masked SPT† | 128 | 4 | 8 | 128 | Mean | 5e-4,1e-3 | 16,16 | 0 | CE |
| Transformer + Masked SPT† | 128 | 4 | 8 | 128 | Mean | 1e-3,1e-3 | 256,128 | 0 | CE |
| Transformer + Rotary + Masked SPT | 128 | 6 | 4 | 512 | Mean | 5e-4,1e-3 | 64,32 | 0 | CE |
| **PathX** | **Features** | **Depth** | **Num Attn. Heads** | **FF size** | **Pooling** | **LR** | **BSZ** | **WD** | **PT Loss** |
| Transformer + Masked SPT† | 128 | 4 | 8 | 128 | Max | 5e-4,5e-4 | 32,8 | 0 | CE |
| Transformer + Masked SPT† | 128 | 4 | 8 | 128 | Max | 5e-4,1e-3 | 256,8 | 0 | CE |
| Transformer + Rotary + Masked SPT | 128 | 5 | 4 | 512 | Max | 5e-4,1e-3 | 32,8 | 0 | CE |
| **SC** | **Features** | **Depth** | **Num Attn. Heads** | **FF size** | **Pooling** | **LR** | **BSZ** | **WD** | **PT Loss** |
| Transformer + Rotary + Masked SPT | 128 | 4 | 4 | 128 | Max | 1e-3,1e-3 | 256,8 | 0 | L1 |
| Transformer + Rotary + Causal SPT | 128 | 4 | 4 | 128 | Max | 1e-3,1e-3 | 256,32 | 0 | L1 |
| **BIDMC** | **Features** | **Depth** | **Num Attn. Heads** | **FF size** | **Pooling** | **LR** | **BSZ** | **WD** | **PT Loss** |
| Transformer - HR + Rotary + Masked SPT | 128 | 4 | 4 | 128 | Max | 5e-4,5e-4 | 32,16 | 0 | L1 |
| Transformer - RR + Rotary + Masked SPT | 128 | 4 | 4 | 128 | Max | 5e-4,5e-4 | 32,16 | 0 | L1 |
| Transformer - SpO2 + Rotary + Masked SPT | 128 | 4 | 4 | 128 | Max | 1e-4,5e-4 | 8,16 | 0 | L1 |

Table 5: Hyper-parameters for SSMs and Pythia in Sections 3.2, 3.3, 3.5. ✗ denotes no evaluation is reported. Values are reported in format (FT - LR, FT - BSZ, PT - LR, PT - BSZ), FT - Fine Tuning (or training on downstream), PT - Pre Training, LR - Learning Rate, BSZ- Batch Size.

| Approach | Listops | Text | Retrieval | Image | Pathfinder | PathX |
|---|---|---|---|---|---|---|
| S4 + Rand Init + SPT | 1e-3,16,1e-3,32 | 1e-3,16,32 | ✗ | 5e-4,64,1e-3,32 | ✗ | 5e-4,16,1e-3,4 |
| S4 + Rand Init | 5e-4,16 | 1e-3,16 | ✗ | 1e-4,128 | ✗ | 5e-4,16 |
| S4 + SPT | 1e-3,16,1e-3,128 | 5e-4,16,5e-4,8 | 5e-4,256,5e-4,8 | 1e-3,512,1e-3,32 | 1e-3,64,1e-3,8 | 1e-3,64,1e-3,8 |
| DLR + Rand Init + SPT | 5e-4,64,5e-4,32 | 5e-4,16,5e-4,8 | ✗ | 1e-3,64,32 | ✗ | 5e-4,16,5e-4,8 |
| DLR + Rand Init | 1e-4,16 | 1e-3,16 | ✗ | 1e-3,16 | ✗ | 5e-4,64 |
| DLR + SPT | 5e-4,16,1e-3,128 | 1e-3,16,5e-4,8 | ✗ | 1e-3,64,1e-3,32 | ✗ | 5e-4,64,5e-4,8 |
| DLR | 5e-4,16 | 5e-4,16 | ✗ | 5e-4,64 | ✗ | 5e-4,64 |
| Pythia | 5e-5,64 | 5e-4,16 | 1e-4,256 | 7e-5,64 | 1e-4,1024 | ✗ |

### C.1.5 PRETRAINING ON TEXT CORPORA

We use the pretrained Pythia model available at: `https://huggingface.co/EleutherAI/pythia-70m-deduped`, and finetune without any regularization. We perform grid search over learning rates and batch sizes, similar to previous sections, with best values listed in Table 6.

### C.1.6 Additional Experiments

For S4, both trained from scratch and pretrained, we follow the hyper-parameters used by Gu et al. (2022a), performing a small grid search similar to previous sections. All Transformers in this section use the Rotary PE method, and use hyper-parameters to match the size of S4, reported in Table 4. On the sCIFAR task Transformers use the same setup as the Image task.

Table 6: hyper-parameters for SSMs in section 3.7. †denotes results are cited from Gu et al. (2022a)

| Approach | SC-Causal | SC-Bi. | sCIFAR | BIDMC-RR | BIDMC-HR | BIDMC-SpO2 |
|---|---|---|---|---|---|---|
| S4 + SPT | 5e-4,16,1e-3,32 | 5e-3,16,1e-3,32 | 5e-4,64,1e-3,32 | 1e-3,64,1e-3,64 | 1e-3,64,1e-3,64 | 1e-3,64,1e-3,64 |
| S4 | 1e-3,16,1e-3,128 | 5e-4,16,5e-4,8 | 5e-4,256,5e-4,8 | 1e-2,32 | 1e-3,32 | 1e-2,32 |

### C.2 The Role of Explicit Priors - Extended Results

We report all results for experiments in section 3.3. The pretrained S4 models with HiPPO initialization are the same as those used in section 3.2.

Table 7: Evaluation of different State Space models with and without SPT.

| Approach | Listops | Text | Image | PathX | Avg. |
|---|---|---|---|---|---|
| S4 + Rand Init + SPT | 61.5 | 90.5 | 87.25 | 91.05 | 82.75 |
| S4 + Rand Init | 59.85 | 88.34 | 75.56 | 73.57 | 74.33 |
| S4 + SPT | 61.25 | 90.34 | 89.36 | 96.94 | 84.47 |
| S4 | 59.60 | 86.82 | 88.65 | 96.35 | 82.85 |
| DLR + Rand Init + SPT | 56.75 | 89.13 | 81.83 | 94.59 | 80.47 |
| DLR + Rand Init | 39.55 | 73.72 | 56.70 | 50.04 | 55.00 |
| DLR + SPT | 60.45 | 89.94 | 87.12 | 96.38 | 83.37 |
| DLR | 57.50 | 79.65 | 79.83 | 92.33 | 77.33 |

### C.3 Additional Experiments - Details

For experiments in this section we follow the scheme described in Section 2 for the experimental setup. Namely, for masked pretraining we use a fixed masking ratio for each task, for SC and BIDMC we use $25\%$ masking and for sCIFAR we use $50\%$. In the SC task, when a causal model is also trained, we pretrain with a causal denoising objective. Due to the significant length of SC sequences, $16K$, we split the input to the attention layer into non-overlapping blocks of size 4000 and allow each block to attend to itself and its neighbour(s), in similar fashion to experiments on PathX.

In BIDMC, we observed that naive finetuning of pretrained models results in very bad performance, which is an artifact of the label and data scales being very different, e.g. common label values for the HR task are $\sim 80$ while input is normalized to $[-1, 1]$. The reported performance is after normalizing the labels to lie in $[-1, 1]$, which does not entail any additional information on the label itself.

## D Compute Requirements of Self Pretraining

In sections 3.1, 3.2 and 3.7 we observed large empirical gains using SPT compared to simply training models from scratch. However, as SPT requires additional compute, it is important to ensure that the reported gains are not an artifact of additional resources and that training longer from scratch does not result in similar gains.

In each of our trained from scratch (TFS) baseline experiments, we ensured that either (1) the training accuracy is almost perfect and validation performance stopped improving for multiple epochs, or

(2) the training loss stopped reducing. E.g. TFS Transformer on PathX fail to exceed 52% training accuracy after 8 epochs (equivalent to 2 days on 4 V100 GPUs), while SPT for 1 day and finetuning for 1 day achieved training accuracy $\geq 78\%$. Across our experiments, we observed case (2) to occur on a small number of runs and case (1) to be the dominant paradigm.

This implies that the gains due to SPT cannot be simply explained by the use of additional compute and possible underfitting of TFS models, but rather by improved generalization as a consequence of the pretraining denoising objective.

To validate this further, we conducted a compute-tied study for the Image and Text tasks from LRA, where we fixed the total number of training epochs across SPT and fineutning (FT) and varied the number of epochs allocated for SPT[5]. As shown in Table 8, even a modest amount of SPT outperforms or closely matches the TFS baseline.

Table 8: Comparison of SPT and trained from scratch (TFS) models in a compute-tied setting. Total number of epochs across SPT and finetuning is fixed and the ratio of epochs dedicated to SPT is varied. Training budget is set to 30 epochs for Text and 150 epochs for Image.

| SPT Epochs | Image | | Text | |
|---|---|---|---|---|
| | Transformer | S4 | Transformer | S4 |
| 0% (TFS) | 75.04 | 87.83 | 79.08 | 87.51 |
| 20% | 84.45 | 87.15 | 90.20 | 89.50 |
| 40 % | 84.95 | 87.72 | 90.56 | 89.10 |
| 60 % | 84.32 | 87.63 | 90.65 | 88.87 |

Despite the efficiency of SPT presented in Table 8, the performance still lags behind the unrestricted setting suggesting. However, in our experiments we observed that the SPT phase reaches close to the peak performance relatively early in the run, and leads to optimization benefits during finetuning, compared to the trained from scratch model. In Figure 5 we provide the training performance for trained from scratch and SPT models, demonstrating the above claims. This suggests that the computational requirements of our evaluation scheme can be potentially reduced, which we leave for future work.

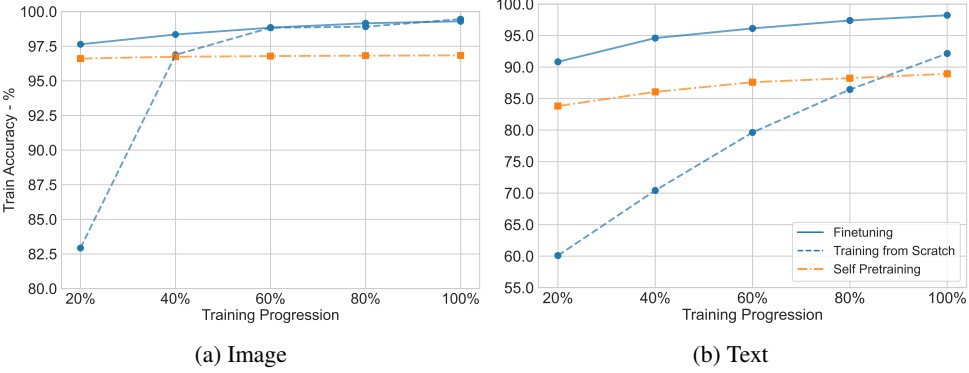

(a) Image                                     (b) Text

Figure 5: Training accuracy on the downstream and denoising task of Transformers on Image and Text from LRA, across epochs, showing the denoising task is solved relatively quickly (orange) and that finetuned models optimize faster (solid vs dashed blue curves). Training budget is set to 30 epochs for Text and 150 epochs for Image.

[5]The performance of S4 shown here is for the model we retrained using the specified number of epochs - different from Table 2 which is cited from the original work.

Table 9: Performance on Image task across model sizes with SPT & trained from scratch.

| Approach | $100K$ | $300K$ | $1M$ | $3M$ | $10M$ |
|---|---|---|---|---|---|
| Transformer+Rotary | 68.51 | 68.51 | 71.50 | 75.04 | 77.88 |
| Transformer+Rotary + Masked SPT | 74.43 | 76.36 | 84.83 | 86.04 | 86.54 |
| S4 | 81.36 | 83.63 | 84.81 | 88.65 | 85.73 |
| S4 + Masked SPT | 83.45 | 86.39 | 88.67 | 89.36 | 88.72 |

# E  SELF PRETRAINING IS EFFECTIVE ACROSS MODEL SCALES

In section 3.4, we demonstrated the effectiveness of SPT across data scales. We now demonstrate that SPT is effective across model sizes as well. We focus on the Image task, which exhibits the largest gap across scales (i.e. the gap between Table 1 and Table 2), and evaluate both S4 and a Transformer with rotary positional embeddings with model sizes spanning four orders of magnitude. While extensive research has been dedicated to study the effect of model scale on the performance of Transformers, less literature is available on scaling-laws for state space models (e.g. S4).[6] The results in Table 9 show that the utility of SPT is maintained across model scales, consistently outperforming the trained from scratch variants.

---

[6]along with model width and depth, state space models have an additional hyperparameter; the state dimension. We found it difficult to scale up the model size without increasing the state size.

