# OpenReview forum: "Never Train from Scratch: Fair Comparison of Long-Sequence Models Requires Data-Driven Priors"
_ICLR.cc/2024/Conference — ICLR 2024 oral_

### Official Review · Reviewer_yToF · 2023-10-23

**Soundness:** 3 good
**Presentation:** 3 good
**Contribution:** 3 good
**Rating:** 8
**Confidence:** 4

**Summary:**

This work questions the common procedure of testing sequence model architectures by directly training on downstream supervised tasks (e.g. the LRA benchmark). They propose to pretrain models on the downstream task data (self pretaining or SPT) before fine-tuning on the task. This significantly closes the gap between many sequence models on long-range sequence tasks. The work also investigates the importance of manually designed biases to help methods capture long-range dependencies and finds that SPT makes these biases less important.

**Strengths:**

- Benchmarks such as Long Range Arena (LRA) are constantly used for new sequence modeling architectures. Questioning some of the assumptions and usefulness of benchmarks such as this is a fresh and interesting direction.

- Most of the experimental results are strong and convincing. The proposed SPT really does seem to improve the performance of methods such as Transformers previously considered to be unable to solve LRA. This is compelling since SPT is more in line with how large models are often trained.

- The investigations of explicit priors and effects across data scales are also interesting angles to explore.

**Weaknesses:**

- The paper does not seem to take the cost of SPT vs training directly on the downstream task into account, or at least does not make this axis of comparison clear.
  - The details are unclear in how much time is spent pretraining vs fine-tuning with the SPT procedure and how this compares to the typical training method for these tasks. These points should be discussed clearly in the main paper.
  - It is stated in Section 3.2 that sub-par LRA performance is often cited as a prime motivating factor for new methods, but it seems efficiency is often just as important a reason new methods are proposed
  - If one method can be directly trained on the task, while another has to first be pretrained and then fine-tuned, one would need to compare the cost/time/compute/etc to achieve a certain performance to determine if one method is superior, at least in many settings (note this point is less relevant in large scale language and vision settings where pretraining is the norm).
  - Further exploration and clarification on these points would improve the paper.

- Even when trained with SPT, it seems from the results that methods such as S4 with structured biases consistently outperform methods with less structured biases across almost every task (I believe Text and Retrieval in Table 2 are the only exceptions to this). The reviewer agrees these differences are not as drastic as it seems when using the traditional procedure, but it still seems the structured biases are helping and the traditional procedure is somewhat predictive of the ordering. Or is this just an artifact of the experiments? A potential discussion on this point could be useful.

- Table 1 lists many efficient attention methods that were originally evaluated on LRA. It would have been interesting to see a couple of these methods also trained with SPT to confirm empirically that they also do not perform as poorly when using SPT. (I suspect this is the case, but currently have to guess since no result is provided).

- The tasks considered in this paper are standard, but nonetheless the addition of less synthetic or larger scale tasks and experiments would also make the paper more compelling to a broader audience

- No code appears to be included (it seems it will be made available based on the anonymized link, but would have been nice to explore during the review).

**Questions:**

1. Could you clarify the pretraining vs fine-tuning procedure and how much time is spent on both for each method? Please let me know if I have missed this in the main paper. Appendix B.1 says models were trained for 200 epochs or 24 h "for pretraining and fine-tuning", but it is unclear if this means 200 epochs of pretraining and 200 epochs of fine-tuning? If so this would seem to be much more time/epochs than some of the baseline methods were trained.

2. In Table 2, was the Transformer + Rotary embeddings trained without SPT? This seems unlikely. Perhaps there is a typo in the description of the Transformer methods in this table?

3. In Table 2, the X "denotes computationally infeasible or unreported results". These are 2 drastically different things and 2 separate symbols should probably be used.

4. The methods evaluated for Figure 2 still seem to use complex valued parameterizations even though they are randomly initialized. Is this still necessary when using SPT? Since complex values can be problematic when scaling large scale systems, it would be interesting if SPT also removed the need for this in SSMs/linear RNNs.

---

> ### Author Response · Authors · 2023-11-17
>
> We are glad the reviewer found our work insightful and inline with common practices when training large models, which was part of the motivating factor for our work, and in general appreciate the detailed response and insightful comments and suggestions. The reviewer raised several questions which we address in order:
> 1. ```The paper does not seem to take the cost of SPT vs training directly on the downstream task into account, or at least does not make this axis of comparison clear.```: This question was raised by all reviewers - please see our unified response above.
> 2. ```It is stated in Section 3.2 that sub-par LRA performance is often cited as a prime motivating factor for new methods, but it seems efficiency is often just as important a reason new methods are proposed```: We agree that SSMs and transformer variants augmented with SSMs are typically more efficient in terms of compute compared to vanilla transformers and **have stated this point on Page 5, para 4**.
> 3. ```...it still seems the structured biases are helping and the traditional procedure is somewhat predictive of the ordering. Or is this just an artifact of the experiments? A potential discussion on this point could be useful.```: We agree with the reviewer that those indeed help and SPT does not render them redundant. Our goal in section 3.3 was to address this exact issue in a controlled setting, evaluating the structured biases in S4, since a comparison directly with Transformers is difficult due to the large differences between models. In the revised version, we included possible implications of different inductive bias between S4 and Transformers as a possible explanation for the remaining gap, mentioned in the beginning of section 3.3, 1st paragraph of page 6.
> 4. ```Table 1 lists many efficient attention methods that were originally evaluated on LRA. It would have been interesting to see a couple of these methods also trained with SPT to confirm empirically that they also do not perform as poorly when using SPT...```: The purpose of our work was to re-examine the common evaluation pipeline for popular long range benchmarks & therefore we chose to focus on widely adopted models and how performance gaps between them change when SPT is incorporated. While we agree this is an interesting point, efficient attention methods are not as widely-used as vanilla attention and hence we leave their evaluation as future work.
> 5. ```The tasks considered in this paper are standard, but nonetheless the addition of less synthetic or larger scale tasks and experiments would also make the paper more compelling to a broader audience```: We agree that to prove the efficiency of SPT in general (rather than only on long sequence tasks) a larger study is in order. In the introduction, 1st paragraph page 2, we cite Krishna et al [1], El-Nouby et al [2] that provide such an analysis on standard NLP and Vision tasks respectively, with further details in the related work section. Along with our work, a large number of tasks and modalities demonstrate the efficacy of SPT. Since we observed training from scratch is widespread when evaluating long range sequence models, we focused on these types of tasks and provide a thorough investigation of SPT and its benefits in that context, and how it leads to a more modern evaluation scheme.
> 6. ```No code appears to be included ...```: We **have added the link to the codebase** made available through the anonymized link at the end of the introduction in page 3.
> 7. ```In Table 2, was the Transformer + Rotary embeddings trained without SPT? This seems unlikely. Perhaps there is a typo in the description of the Transformer methods in this table?```: In Table 2 the first line (Transformer + Rotary) is indeed trained from scratch, the difference between this model and in Table 1 is the use of Rotary embeddings, as well as the larger model size for several tasks.
> 8. ```In Table 2, the X "denotes computationally infeasible or unreported results. These are 2 drastically different things and 2 separate symbols should probably be used.```: As per your suggestion we **have fixed** this in the revised version.
> 9. ```The methods evaluated for Figure 2 still seem to use complex valued parameterizations even though they are randomly initialized. Is this still necessary when using SPT? Since complex values can be problematic when scaling large scale systems, it would be interesting if SPT also removed the need for this in SSMs/linear RNNs.```: We agree with the reviewer this is an interesting experiment, but leave it for future work.
>
> [1] Kundan Krishna, Saurabh Garg, Jeffrey Bigham, and Zachary Lipton. “Downstream datasets make surprisingly good pretraining corpora”.
> [2] Alaaeldin El-Nouby, Gautier Izacard, Hugo Touvron, Ivan Laptev, Herv  ́e Jegou, and Edouard Grave. Are large-scale datasets necessary for self-supervised pre-training?

---

> > ### Comment · Reviewer_yToF · 2023-11-21
> >
> > Thank you for the additional details and clarifications. I think this is a good paper that will be of interest to the community and I have increased my score.

---

> > > ### Author Response · Authors · 2023-11-21
> > >
> > > We thank the reviewer for the insightful feedback and for increasing the score! Please dont hesitate to let us know in case of any further questions.

---

> > > > ### Public Comment · ~Mohamed_Al_Hajeri1 · 2026-06-22
> > > >
> > > > Your SPT finding addresses evaluation fairness at initialization. I want to add a structural observation from empirical work at the weight geometry level.
> > > > When models are trained across GPU nodes — tensor or pipeline parallelism — each fragment of the same MLP layer trains on a different machine with different data batches, different gradient trajectories, and different numerical conditions. Stitched back, it presents as one weight but carries multiple training histories. The LM head reads this as unified signal. It is not. This fragmentation, I believe, is a structural contributor to hallucination — not confusion about the world, but incoherence between the fragments of the model's own weights.
> > > > I ran 17 iterations attempting MLP weight projection from seven donors into a calibrated 458M host. Every variant failed. The barrier is the host LM head's joint training with its native MLPs — it creates a basis-specific reading expectation that any foreign or fragmented signal violates regardless of projection quality.
> > > > The mathematics of projection are sound. The premise that weights trained separately can be unified after the fact does not survive empirical testing.
> > > > Full empirical data: https://zenodo.org/records/20446689
> > > >
> > > > DOI: 10.5281/zenodo.20446689

---

### Official Review · Reviewer_7n9E · 2023-10-31

**Soundness:** 3 good
**Presentation:** 3 good
**Contribution:** 3 good
**Rating:** 8
**Confidence:** 4

**Summary:**

This paper studies the effectiveness of self-training (SPT) using its own downstream data with Transformer and state space models (in particular S4) for long-range sequence modeling. Specifically, the studies include the impact of SPT on Transformer, S4, and Diagonal Linear RNNs for long-range sequences. Also, the effect of SPT across data size and data-driven vs HiPPO kernel initialization are analyzed. The experiments show that SPT overall improves the performance on Long Range Arena benchmarks, speech commands, CIFAR, and some regression tasks.

**Strengths:**

- The paper is clearly written.
- The experimental setup sounds and the results are informative.
- The analysis of conv. kernels learned via SPT compared to the HiPPO kernels is novel and interesting.

**Weaknesses:**

- Most experiments are performed on Long Range Arena which is relatively small or synthetic.

- The main self-pretraining results (Tables 1 and 2) are not with the latest Transformers and SSMs.

- Some experimental analysis is lacking. See my questions below.

**Questions:**

- Self-training may be effective not only across the data scale but also the model size. Have you tried the same experiments with different model sizes? It would be interesting to see the phenomenon.

- It looks like the hybrid models like SPADE and MEGA outperform other models including transformer+SPT and S4+SPT for many Long Range Arena tasks. Would SPADE+SPT or MEGA+SPT further improve the performance? Is there a reason this comparison is not included in the paper?

- Regarding the experiment about pretraining on text corpora: I appreciate the idea of comparing results with a pretrained model on a large language dataset. However, the current results are meaningless since the downstream task datasets are very different from Pythia 70M. I'm not sure we are able to find the right dataset to cover all the tasks for Long Range Arena. What about separating this experiment from Long Range Arena and showing the comparison for another downstream task on the language domain?

- Are S4/Transformers trained with the same number of epochs as S4+SPT/Transformers+SPT (including pretraining+finetuning)? If it's not, these results should be added. I'm asking to make sure the single-trained models are not undertrained.

---

> ### Author Response · Authors · 2023-11-17
>
> We appreciate your response on finding our work thorough and informative, specifically the remark on our analysis for conv. kernels between HiPPO and SPT models. The reviewer raises several important questions that we address in order:
> 1. ```Self-training may be effective not only across the data scale but also the model size```: This observation indeed aligns with our reported results. Our experiments show that original model sizes used in LRA visual tasks (Image, Pathfinder, PathX) are too small for high performance (Table 1) which led us to scale up the model sizes (Table 2), as mentioned in the 1st paragraph of page 5, section 3.2. To further investigate this, **we plan on adding a scaling experiment as suggested in the next revision**, comparing performance across model sizes for SPT and trained from scratch models.
> 2. ```Would SPADE+SPT or MEGA+SPT further improve the performance? Is there a reason this comparison is not included in the paper?```: The reported results for hybrid models indeed outperform SPT variants, in our work we focus on transformers and S4 as they seem to be the most widely-used. We note that we did try training MEGA from scratch but were unable to reproduce the reported results.
> 3. ```The current results are meaningless since the downstream task datasets are very different from Pythia 70M. I'm not sure we are able to find the right dataset to cover all the tasks for Long Range Arena. What about separating this experiment from Long Range Arena and showing the comparison for another downstream task on the language domain?```: The purpose of Pythia experiment is to see if pretraining on text is helpful on LRA, as has been observed on various modalities such as molecular data, tabular data, code and speech [1,2,3]. The issue of finding a unifying dataset that can be used for pretraining across modalities seen in LRA is indeed difficult, as mentioned by the reviewer, and we view SPT as a remedy for it.  For text-based tasks specifically, the question of evaluating the effectiveness of SPT vs. pretraining on large text corpora is discussed thoroughly in Krishna et. al. [4], showing that in many cases SPT rivals PT on large corpora as mentioned both in the 1st paragraph of page 2, and the related work section. Due to their comprehensive analysis, we focused on SPT in the context of evaluating models on long-range benchmarks. Nonetheless, for a more reliable comparison **we have added an additional baseline of **Pythia with random initialization** in the revision**, denoted as Pythia 70M (Rand Init), and **expanded the discussion in section 3.5** following these addition.  The randomly initialized baseline from which clearly shows the benefits of text pretraining are more easily observed. Yet in some cases, the pretrained model fails to significantly outperform the randomly initialized model, unlike SPT, which provides benefits across the board.
> 4. ```Are S4/Transformers trained with the same number of epochs as S4+SPT/Transformers+SPT```: This question was raised by all reviewers - please see our unified response above.
>
> [1] Igor Melnyk, Vijil Chenthamarakshan, Pin-Yu Chen, Payel Das, Amit Dhurandhar, Inkit Padhi, Devleena Das, Reprogramming Pretrained Language Models for Antibody Sequence Infilling.
> [2] Herzig et al., TAPAS: Weakly Supervised Table Parsing via Pre-training.
> [3] Ao et al., SpeechT5: Unified-Modal Encoder-Decoder Pre-Training for Spoken Language Processing.
> [4] Kundan Krishna, Saurabh Garg, Jeffrey Bigham, and Zachary Lipton. “Downstream datasets make surprisingly good pretraining corpora”.

---

> > ### Author Response · Authors · 2023-11-20
> >
> > Following the reviewers suggestion, we have added a scaling experiment across model sizes in Appendix E. The results show that SPT is indeed effective across multiple scales, showing clear benefits for both Transformers and S4.
> >
> >
> > ## Performance on Image task across model sizes with SPT & trained from scratch
> >
> > | Approach                          | 100K  | 300K  | 1M    | 3M    | 10M   |
> > |-----------------------------------|-------|-------|-------|-------|-------|
> > | Transformer+Rotary                | 68.51 | 68.51 | 71.50 | 75.04 | 77.88 |
> > | Transformer+Rotary + Masked SPT   | 74.43 | 76.36 | 84.83 | 86.04 | 86.54 |
> > | S4                                | 81.36 | 83.63 | 84.81 | 88.65 | 85.73 |
> > | S4 + Masked SPT                   | 83.45 | 86.39 | 88.67 | 89.36 | 88.72 |

---

> > > ### Comment · Reviewer_7n9E · 2023-11-22
> > > **response to authors**
> > >
> > > Thank you for the clarification and the additional experiments. They are very helpful, especially the experiments with compute-tied setting and the comparison across the model size.
> > >
> > > Also, the Pythia experiment with random init is useful. However, after seeing the Pythia70M with random init result, I wonder if the low performance is due to other factors, e.g., architecture, embeddings, etc, or needs some other tricks for training the large model. To see the impact of pertaining on text corpora vs SPT, it would be good to add either (1) Transformers + Rotary with pertaining on text or (2) Pythia70M+Masked SPT. (1) would be more helpful but takes more effort, so (2) would be a good alternative. I suggest adding one of these experiments in the final version.
> > >
> > > Overall, it is an interesting paper and has great insight. I increase my score.

---

> > > > ### Author Response · Authors · 2023-11-22
> > > >
> > > > We thank the reviewer for his remarks and constructive feedback as well as increasing the score.
> > > >
> > > > We agree that investigating SPT with Pythia is interesting - we will consider adding it to the final version.

---

### Official Review · Reviewer_fh2t · 2023-10-31

**Soundness:** 3 good
**Presentation:** 3 good
**Contribution:** 3 good
**Rating:** 8
**Confidence:** 4

**Summary:**

This paper provides a suite of experiments to show that self pretraining (SPT), i.e., pretraining with denoising objectives on only downstream data, most often closes the performance gap between Transformers and state space models (SSMs) on the Long Range Arena benchmark. In the case of Transformers the performance gains from the incorporation of SPT range from 8 to 15% across tasks.

The experiments also show that in the case of SSMs, manually-designed biases become increasingly redundant when SPT is incorporated.

More generally, the results suggest the evaluation of different architectures on supervised tasks should incorporate SPT for reliable performance estimation.

**Strengths:**

**S1.** The presentation of the main ideas, related work and experimental results is clear.

**S2.** The incorporation of SPT is efficient and extremely effective compared to only training from scratch.

**S3.** The experimental results are thorough and support the main claims in the paper.

**Weaknesses:**

**W1.** There are no results on computing requirements for SPT and, e.g., how to best combine SPT with supervised fine-tuning.

 **W2.** The results on PathX-256 suggest SPT failed to close the gap in this case. This seems to warrant further investigation.

**Questions:**

**Q1.** What can be said about the results of Transformers and S4 on PathX-256?

 **Q2.** What is the point of the experiment with Pythia? For instance, what is the single Pythia row in Table 2 to be compared with?

---

> ### Author Response · Authors · 2023-11-17
>
> We are encouraged to hear that the reviewer found our work to be thorough and our method effective. The reviewer raises several important questions that we address in order:
> 1. ```how to best combine SPT with supervised fine-tuning```: This question was raised by all reviewers - please see our unified response above.
> 2. ```results of Transformers and S4 on PathX-256```:  The question about PathX-256 results from a misuse of notation. We could not fit a single sample on PathX-256 (input length 65K) on our 24GB GPU with Transformer with chunked attention and therefore could not evaluate them at all. We have **modified the notation** in Table 2 to clarify the difference between setups that are computationally infeasible and unreported results.
> 3. ```What is the point of the experiment with Pythia?...```: We thank the reviewer for pointing to the lack of clarity in the Pythia section. It is common practice in various areas of ML to take LLMs pretrained on text and directly adapt them to other modalities such as molecular data, tabular data, code and speech - researchers have reported significant benefits from this approach [1, 2, 3]. In the Pythia experiments, we wanted to examine if pretraining on a large text corpus would result in large gains on LRA or whether maintaining the modality between PT and FT tasks is important. We agree that comparing the Pythia results to the available Transformer results makes comparison difficult due to different architecture and model sizes. Therefore, in the revised version, **we have added an additional baseline of Pythia with random initialization**, denoted as Pythia 70M (Rand Init) and indeed observe benefits from pretraining on text, yet not on all tasks, which shows the importance of SPT leading to to better performance across the entire suite.
>
> [1] Igor Melnyk, Vijil Chenthamarakshan, Pin-Yu Chen, Payel Das, Amit Dhurandhar, Inkit Padhi, Devleena Das, Reprogramming Pretrained Language Models for Antibody Sequence Infilling.
> [2] Herzig et al., TAPAS: Weakly Supervised Table Parsing via Pre-training.
> [3] Ao et al., SpeechT5: Unified-Modal Encoder-Decoder Pre-Training for
> Spoken Language Processing.

---

> > ### Comment · Reviewer_fh2t · 2023-11-21
> > **Response to authors**
> >
> > Thank you for the clarifications. The additional detail and experimental results on the combination of SPT and SFT (and scaling) are very helpful.
> >
> > I am increasing my score.

---

> > > ### Author Response · Authors · 2023-11-21
> > >
> > > We kindly thank the reviewer for the fruitful discussion and increasing the score. Please let us know in case of any additional questions.

---

### Official Review · Reviewer_YbGG · 2023-11-01

**Soundness:** 4 excellent
**Presentation:** 3 good
**Contribution:** 3 good
**Rating:** 8
**Confidence:** 3

**Summary:**

The paper demonstrates that random initialization of model weights on long sequence benchmarks leads to severe underestimation in performance of transformer architectures. The results show that pre-training on the training data with autoregressive/masked prediction objectives results in much better initializations and final performance. With self pre-training the performance gap between state space models specifically designed for handling long sequences and traditional transformers is much smaller than shown in prior work. Moreover, the self pre-training improves the performance on state space models as well. Overall the paper points out an important baseline which should be adopted more broadly when evaluating different architectures for long sequence tasks.

**Strengths:**

* The paper does a fairly through evaluation of SPT and its impact on evaluation on various benchmarks. The evaluation clearly shows gaps in the current evaluation of different architectures on long sequence tasks.
* In addition to providing guidance on evaluation practices the experiments also show the effectiveness of data driven initialization across both transformers and state space models. It is also interesting to see SPT to provide better initialization at smaller data scales for state space models.
* The paper also demonstrates that simplified state space models can perform competitively to their complicated counterparts when initialized with SPT

**Weaknesses:**

* The paper largely compares the different models on the effectiveness in terms of benchmark accuracy. It would be good to include commentary on SPT computational costs relative to initializations of state space models.

**Questions:**

* Figure 3, in the smallest data setting it seems like SPT does not provide monotonically increasing gains as the data size reduces. Is there guidance on what scale of data SPT ends up resulting in poor initializations that the ones used in state space models.

---

> ### Author Response · Authors · 2023-11-17
>
> We thank the reviewer for the supportive review and address the raised weakness and questions as follows:
> 1. ```SPT computational costs relative to initializations of state space models```: This question was raised by all reviewers - please see our unified response above.
> 2. ```... on what scale of data SPT ends up resulting in poor initializations that the ones used in state space models``` :  We clarify that the plots in Figure 3 show **relative gains** offered by SPT over trained from scratch baseline and for performing SPT standard S4 initialization was used. At extremely low data regimes (0.5% of the original data), the absolute model performance after finetuning on the downstream task is low and hence the relative gain offered by SPT is not significant - however it is not negative, i.e., **not inferior to the trained from-scratch baseline**. 0.5% of the original data amounts to 225 samples for the Image task and 125 samples for the Text task. As a general guidance, when performance on the pretraining task itself approaches random accuracy on the validation set, we would expect SPT not to be beneficial, but even for 125 samples in Text task, pre-training accuracy is ~ 50% which is significantly higher than chance performance for character-level language modeling. We have added the original dataset sizes to the caption of Figure 3.

---

> > ### Comment · Reviewer_YbGG · 2023-11-22
> > **Response to the authors**
> >
> > Thank you for the clarifications and the updates to the paper clarifying the computational aspects. This will be an informative paper for the community.

---

> > > ### Author Response · Authors · 2023-11-22
> > >
> > > We thank the reviewer for a productive discussion and his comments about our work.

---

### Author Response · Authors · 2023-11-17
**Computational cost of Self Pretraining**

We sincerely thank all reviewers for raising questions about computational cost of SPT. Our training setup, as listed in appendix B.1, was pretraining for either 200 epochs or 24h, first of the two, and finetuning similarly. The only exceptions are PathX and Speech Commands with Transformers, pretrained for 24h and finetuned for up to 5 days. **In the updated revision, we clearly refer to the relevant appendix for compute details** for better clarity.

The computational overhead of SPT has 2 aspects. First is possible undertraining while training from scratch, on the vast majority of the tasks we have validated that training accuracy is almost 100% and validation performance no longer increases for multiple epochs, **which is explicitly mentioned in paragraph 5 page 5, section 3.2 in the updated revision**. Rare exceptions are (1) runs where training performance stopped improving or did not improve at all, e.g. on PathX, (2) when the hyperparameter search led to undertrained model performing better on the validation set.  Hence, **training from scratch for longer will not improve performance on the downstream task**, as the issue is generalization rather than optimization.

The 2nd aspect is **evaluation in a compute-tied setting**, in the revised version we **added a study in appendix D** of SPT against training from scratch in which the **total number epochs** is fixed, and the epochs allocated to SPT and finetuning are varied. Our results show the benefits of SPT are robust to a setting of restricted compute, for S4 and Transformers alike. Furthermore, the results show a small fraction of epochs for SPT suffices for significant performance gains, pointing to fast optimization of the pretraining objective, this subject is further analyzed showing SPT both optimizes fast, reaching close to peak performance early in training, and leads to faster optimization on the downstream task compared to the trained from scratch model. **An additional paragraph is added to the end of section 3.2 in page 5, highlighting the results in the appendix and the additional experiments.**

## Comparison of SPT and trained from scratch (TFS) models in a compute-tied setting
Total number of epochs across SPT and finetuning is fixed and the ratio of epochs dedicated to SPT is varied. Training budget is set to 30 epochs for Text task and 150 epochs for Image task.
| SPT Epochs | Image (Transformer) | Image (S4) | Text (Transformer) | Text (S4) |
|------------|---------------------|------------|--------------------|-----------|
| 0% (TFS)   | 75.04               | 87.83      | 79.08              | 87.51     |
| 20%        | 84.45               | 87.15      | 90.20              | 89.50     |
| 40%        | 84.95               | 87.72      | 90.56              | 89.10     |
| 60%        | 84.32               | 87.63      | 90.65              | 88.87     |

---

### Meta-Review · Area_Chair_1Cbb · 2023-12-06

**Metareview:**

This paper compares transformer models versus state space models on long-range sequential data. Unlike prior work, which has primarily considered trained-from-scratch transformers, this work demonstrates that transformers can significantly outperform state space models when they are first pretrained on downstream task data (self pretraining, or SPT). Overall, the paper is clearly written, and the experiments are broad and well executed. This paper presents clear evidence in support of their hypothesis; the level of improvement under pretraining is quite significant and will be of sure interest to the community. Therefore, I recommend that this paper is accepted.

**Justification For Why Not Higher Score:**

N/A

**Justification For Why Not Lower Score:**

This paper has surprising and significant results on a timely topic. The experiments are thorough and well executed, and the paper is clearly written. It is exemplary of a well-executed ICLR paper, and it will be of interest to the broader community.

---

### Decision · Program_Chairs · 2024-01-16

Accept (oral)